

# Effects of number of parallel runs and frequency of bias-strength replacement in generalized ensemble molecular dynamics simulations

Takuya Shimato[1],*, Kota Kasahara[2],*, Junichi Higo[3] and Takuya Takahashi[2]

[1] Graduate School of Life Sciences, Ritsumeikan University, Kusatsu, Shiga, Japan
[2] College of Life Sciences, Ritsumeikan University, Kusatsu, Shiga, Japan
[3] Graduate School of Simulation Studies, University of Hyogo, Kobe, Hyogo, Japan
* These authors contributed equally to this work.

## ABSTRACT

**Background:** The generalized ensemble approach with the molecular dynamics (MD) method has been widely utilized. This approach usually has two features. (i) A bias potential, whose strength is replaced during a simulation, is applied. (ii) Sampling can be performed by many parallel runs of simulations. Although the frequency of the bias-strength replacement and the number of parallel runs can be adjusted, the effects of these settings on the resultant ensemble remain unclear.
**Method:** In this study, we performed multicanonical MD simulations for a foldable mini-protein (Trp-cage) and two unstructured peptides (8- and 20-residue poly-glutamic acids) with various settings.
**Results:** As a result, running many short simulations yielded robust results for the Trp-cage model. Regarding the frequency of the bias-potential replacement, although using a high frequency enhanced the traversals in the potential energy space, it did not promote conformational changes in all the systems.

## INTRODUCTION

In the past several decades, the molecular dynamics (MD) method has been widely applied to investigate the microscopic behavior of molecular systems. Although advances in high-performance computing technology have extended the timescale that is reachable by MD simulations (*Salomon-Ferrer et al., 2013*; *Shaw et al., 2014*; *Abraham et al., 2015*), there is still a large gap from experimental measurements. In particular, it is not straightforward to characterize the free-energy landscape (FEL) of a complex molecular system, because the characteristics of conformational ensembles obtained via canonical MD simulations largely depend on the initial conditions. To solve this problem, the generalized ensemble (GE) approach has been extensively developed and applied to the MD method. The GE approach enhances the conformational sampling using some tricks. First, in many GE methods, the conformational sampling can be performed with many

Corresponding author
Kota Kasahara,
ktkshr@fc.ritsumei.ac.jp

parallel runs of simulations in a coupled or independent manner. For example, the replica-exchange MD (REMD) method (*Sugita & Okamoto, 1999*) involves performing many simulations of the same system, i.e., replicas, with different temperatures. The replicas with adjacent temperatures are coupled by exchanging their temperatures via Monte Carlo trials. On the other hand, the multicanonical MD (McMD) method (*Nakajima, Nakamura & Kidera, 1997*) can be performed by multiple independent runs, and a resultant ensemble is obtained by concatenating the trajectories of these runs (*Ikebe et al., 2010*). Second, the GE approach generates a non-Boltzmann distribution by applying bias potential, e.g., heating/cooling in the entire system or a part of the system, scaling the potential energies, and applying spring potentials for parts of system. These biases enhance the conformational changes of molecules and avoid trapping the molecular system at local minima in the FEL. During a simulation, the strength of the bias is frequently replaced, and the system alternates between different bias conditions. After simulations, a canonical ensemble can be obtained by reweighting each snapshot in the sampled conformational ensemble (*Souaille & Roux, 2001*; *Shirts & Chodera, 2008*).

For using these two features, users must adjust some settings. First, the number of runs is an adjustable parameter. In the case of the REMD method, using a larger number of replicas allows wider overlaps of the energy distributions between adjacent replicas and results in a higher acceptance probability. However, increasing the number of runs proportionally increases the computational costs. Users must choose the optimal balance between the number of runs and the length of each run according to the available computational resources. Previously, *Ikebe et al. (2010)* reported that an increase in the number of independent runs of McMD yields efficient exploration of a wider area of the conformational space. However, the balance between the number of runs and the length of each run has not been discussed. Second, the frequency of the bias-strength replacement is also adjustable. In the REMD method, the frequency of replica-exchange trials must be specified by users. Other methods using a continuous bias strength, e.g., McMD and adaptive umbrella sampling (AUS), can control the frequency of bias-strength replacement by using the virtual-system coupling scheme (*Higo, Umezawa & Nakamura, 2013*; *Higo et al., 2015*), as described later. It is reported that the frequency of the bias-strength replacement affects the resultant ensembles for the REMD method (*Periole & Mark, 2007*; *Sindhikara, Meng & Roitberg, 2008*; *Rosta & Hummer, 2009*; *Sindhikara, Emerson & Roitberg, 2010*; *Jani, Sonavane & Joshi, 2014*; *Iwai, Kasahara & Takahashi, 2018*). Although higher frequencies enhance the traversals in the temperature space, they are suspected as an origin of artifacts. Although the effects of these features have been examined, these studies were mainly based on simple model peptides with helix–coil transitions. The effects of the features for more practical cases, e.g., a protein folding–unfolding transition, are not fully understood. More importantly, the relationship between these effects and the complexities of molecular systems, e.g., the degree of freedom and ruggedness of the FEL, are expected to be revealed.

In this study, we aim to elucidate the effects of the number of runs and the bias-replacing frequency for the GE method on the resultant conformational ensembles of molecular models including a foldable mini-protein and disordered model peptides.

We utilized the trivial-trajectory parallelized virtual-system coupled McMD (TTP-V-McMD) method (*Ikebe et al., 2010*; *Higo, Umezawa & Nakamura, 2013*), which is a variant of the McMD method, for simulating the three molecular models with an explicit solvent: (i) Trp-cage, (ii) 8-residue poly-glutamic acid (PGA8), and (iii) 20-residue poly-glutamic acid (PGA20). We chose these models as test cases to examine the simulation conditions because they are sufficiently small for elucidating their conformational ensembles within a practical computational time in addition to the fact that their structural properties have been well studied thus far. Trp-cage, which is a mini-protein consisting of 20 amino acids, has been widely studied as a prominent model of protein folding (*Ahmed et al., 2005*; *Hudáky et al., 2007*; *Hałabis et al., 2012*). Poly-glutamic acids have been used as model peptides to characterize the conformational properties of polypeptides (*Clarke et al., 1999*; *Kimura et al., 2002*; *Finke et al., 2007*; *Donten & Hamm, 2013*; *Ogasawara et al., 2018*). We analyzed their FELs under various parameter settings to provide a guide for adjusting these parameters for the GE methods. The questions to be answered are as follows: (1) Which condition is more efficient: many short simulations or a small number of long-term simulations? (2) Which is better: frequent or less frequent replacement of the bias strength? Moreover, we discuss the relationship between the relaxation of the energy and that of the protein conformation. While the McMD method enhances the relaxation in the energy space, it is not guaranteed to enhance the relaxation in the conformational space. We analyzed these two relaxation processes using the McMD trajectories calculated with the various settings.

## MATERIALS AND METHODS

We calculated the FELs of the three explicitly solvated molecular models: Trp-cage, PGA8, and PGA20, by using the TTP-V-McMD method with various settings. The theory of McMD, virtual-system coupled McMD (V-McMD), and trivial-trajectory parallelization (TTP) is briefly presented in the following subsections. Then, the simulation protocol applied in this study is described.

### Multicanonical MD

The McMD method efficiently explores the conformational space of a molecular system, by applying a biasing energy term. The Hamiltonian $H$ of the system is

$$H = K + E_{\mathrm{mc}}, \tag{1}$$

where $K$ and $E_{\mathrm{mc}}$ denote the kinetic energy and multicanonical energy, respectively. $E_{\mathrm{mc}}$ is defined as follows:

$$E_{\mathrm{mc}} = E + RT \ln \mathrm{P_c}(E, T), \tag{2}$$

where $E$ is the potential energy, and the second term corresponds to the bias potential. $R$ is the gas constant, and $\mathrm{P_c}(E, T)$ denotes the canonical distribution at the temperature $T$:

$$\mathrm{P_c}(E, T) = \frac{\mathrm{n}(E) \exp\left(-\dfrac{E}{RT}\right)}{Z_{\mathrm{c}}(T)}, \tag{3}$$

where n($E$) denotes the density of states, and $Z_c(T)$ is the partition function of the canonical distribution at the temperature $T$. With this definition, the potential energy distribution of an ensemble obtained from the McMD, or the multicanonical distribution, $P_{mc}(E)$, becomes uniform:

$$P_{mc}(E) = \frac{n(E) \exp\left(-\dfrac{E_{mc}}{RT}\right)}{Z_{mc}(T)}$$

$$= \frac{n(E) \exp\left(-\dfrac{E}{RT}\right)}{P_c(E, T) \, Z_{mc}(T)} = \frac{Z_c(T)}{Z_{mc}(T)} = const. \tag{4}$$

As a result, the McMD method performs a random walk in the potential energy space and generates a uniform distribution of potential energy in a resultant ensemble. After a multicanonical ensemble is obtained, a canonical ensemble at any temperature in a sampled energy range can be generated by reweighting the probability of existence of each snapshot.

Equations (3) and (4) include an analytical form of n($E$), which is usually unknown a priori. Therefore, n($E$) is approximated as a parametric function, e.g., the polynomial function, and its parameters are estimated by iterations of McMD simulations to make $P_{mc}(E)$ near-uniform. In the $i$th iteration, the bias potential is calculated using Eq. (2) with the canonical distribution obtained from the $(i-1)$th iteration, i.e., $P_c^{i-1}(E, T)$. As the result of the $i$th iteration, we obtain $P_{mc}^i(E)$. $P_c^i(E, T)$ can be calculated as

$$P_c^i(E, T) = P_{mc}^i(E)P_c^{i-1}(E, T). \tag{5}$$

See Higo et al. (2012) for details.

## Virtual-system coupled McMD

Virtual-system coupled McMD (V-McMD) introduces a virtual system, which interacts with the molecular system, and the multicanonical ensemble is calculated for the entire system consisting of these two subsystems (Higo, Umezawa & Nakamura, 2013). In practice, this method can be roughly interpreted as a combination of McMD and the simulated tempering method. The simulated tempering method replaces the system temperature with the Metropolis criterion and performs a canonical simulation until the next replacement trial. On the other hand, in V-McMD, the potential energy space is split into several regions (Fig. S1), and the molecular system is trapped in one of these regions. With a certain time interval ($t_{VST}$), the molecular system replaces the region to be trapped. The state variable governing which region traps the molecular system is called the "virtual state," and the system defined by the virtual state is called the "virtual system." The energy range of each virtual state is defined to be overlapped with the adjacent virtual states. When the molecular system has the potential energy $E_k$ in the overlapped region of the $i$th and $(i + 1)$th virtual states, the state transition between these two virtual

states can occur. Because this transition does not change the atomic coordinates or potential energy, the Metropolis criterion of this state transition is always satisfied. The time interval of virtual-state transitions ($t_{VST}$) should be determined arbitrarily by users. See *Higo, Umezawa & Nakamura (2013)* for details.

## Trivial-trajectory parallelization

According to the theory of TTP, trajectories of multiple independent McMD runs with the same molecular system and different initial conditions can be treated as a single trajectory of an McMD simulation by concatenating the trajectories in an arbitrary order. This theory requires the condition that the initial coordinates of each run are sampled from the multicanonical distribution. Because the initial coordinates of production runs can be obtained from the near-uniform potential energy distribution generated by iterative simulations, it is expected that this condition holds. The McMD method with the TTP theory, which is called the TTP–McMD method, can be considered as a hybrid Monte Carlo sampler, by assuming that the system transitions from the last snapshot of the $i$th run (the microscopic state $m_{il}$) to the first snapshot of the $j$th run (the microscopic state $m_{jf}$) via a Monte Carlo step (Fig. S2). See *Ikebe et al. (2010)* for details.

## Simulation protocol

We studied the three molecular systems, which are Trp-cage, PGA8, and PGA20 in an explicitly solvated cubic periodic boundary cell, by using the TTP-V-McMD method. Random coil structures of Trp-cage, PGA8, and PGA20 were constructed using the Modeller software (*Webb & Sali, 2016*) without any template. The termini of the PGAs were capped with acetyl and N-methyl groups, and the termini of the Trp-cage were not capped. Each of these molecular models was plased into a cubic box filled by water molecules; the number of water molecules were 5,097, 2,879, and 3,800 for Trp-cage, PGA8, and PGA20, respectively. In addition, a $Cl^-$ ion was added to the Trp-cage model to cancel the net charge of the system. The net charge of the PGA models was zero because all the Glu residues were protonated.

The system was relaxed by using the GROMACS software (*Pronk et al., 2013*). Energy minimizations were successively applied using the steepest descent and conjugate gradient methods. Then, an MD simulation under a constant-pressure ensemble with the Berendsen barostat was performed for 1 ns. In the first half of the simulation, gradual heating from 10 to 300 K was applied. In the simulation, the positions of the heavy atoms of the Trp-cage were restrained, the bond lengths were not constrained, and the integration time step ($\Delta t$) was 0.5 fs. Subsequently, an additional constant-pressure relaxation was applied for 1 ns with $\Delta t$ = 2.0 fs, and the covalent bonds to hydrogen atoms were constrained using the LINCS method (*Hess et al., 1997*; *Hess, 2008*). The final configuration of each model was used for the TTP-V-McMD simulations. The cell dimensions of these configurations were 54.0378, 44.6116, and 49.1174 Å for Trp-cage, PGA8, and PGA20, respectively.

For each model, the following steps were performed using our MD simulation program, which is called myPresto/omegagene and is tailored for GE simulations

(*Kasahara et al., 2016*). The protein conformation was randomized with a constant-temperature simulation at 800 K. By using 30 snapshots taken from a trajectory with an interval of 300 ps, 30 independent runs were simulated with a gradual decrease in the temperature from 629 to 296 K to estimate the density of states. Successively, the TTP-V-McMD simulations were iteratively performed while updating the estimation of the density of states (*Higo, Umezawa & Nakamura, 2013*). A total of 84 production runs were performed ($N_{run}$ = 84) for each of three different interval times for the virtual-state transitions ($t_{VST}$) meaning the interval times for bias-potential replacement: $t_{VST}$ = 0.002, $t_{VST}$ = 0.2, and $t_{VST}$ = 20 ps. The simulation length of each run ($t_{run}$) was 50 ns except for the Trp-cage model with $t_{VST}$ = 0.2 ps, $t_{run}$, of which the simulation length was 200 ns. In total, 50.4 µs of trajectories were simulated as production runs. The virtual system was divided into seven states that cover the energy range corresponding to the canonical distribution from 296 to 629 K. The velocity scaling method (*Berendsen et al., 1984*) was applied to maintain the system temperature.

For the potential parameters, the AMBER ff99SB-ILDN force field (*Lindorff-Larsen et al., 2010*), the ion parameter presented by *Joung & Cheatham (2008)*, and the TIP3P water model (*Jorgensen et al., 1983*) were applied. The electrostatic potential was calculated using the zero-multipole summation method, which is a non-Ewald scheme (*Fukuda, 2013*; *Fukuda, Kamiya & Nakamura, 2014*). The zero-dipole condition with the damping factor α = 0 was used (*Fukuda, Yonezawa & Nakamura, 2011*; *Fukuda et al., 2012*).

## Comparison of simulated ensembles among different settings

On the basis of the trajectories obtained from of the TTP-V-McMD production runs, the effects of the simulation conditions, i.e., the time interval for bias-strength replacement ($t_{VST}$), the number of independent runs ($N_{run}$), and the simulation time of each run ($t_{run}$), were assessed.

For the Trp-cage model, we analyzed the FEL for various conformational ensembles on the basis of the two structural parameters: the root-mean-square deviation (RMSD) of Cα atoms from the native conformation (PDB ID: 1L2Y, model 1), which is denoted as $RMSD_{native}$, and the radius of gyration ($R_g$). The FEL is visualized as the map of the potential of mean forces (PMF) on the plane defined by these two parameters. We defined the *reference* ensemble as the ensemble calculated for the conditions of $t_{run}$ = 200 ns, $N_{run}$ = 84, and $t_{VST}$ = 0.2 ps, because it is expected to have the highest reliability owing to its abundance of samples (it comprises a total of 16.8 µs of simulations). The FELs analyzed in various conditions were compared with the reference FEL with regard to the Pearson correlation coefficient of the PMF ($PCC_{PMF}$). To calculate the $PCC_{PMF}$ for a pair of FELs, bins without samples in one of the two FELs were ignored. In addition, the probability of the existence of the native conformations in each ensemble ($P_{native}$) was measured to characterize each ensemble. The native conformations are defined as the conformations with $RMSD_{native} \leq 2.0$ Å.

For the PGA models, the FELs were analyzed using principal component analysis (PCA) based on the Cα–Cα distances (28 and 190 dimensions for PGA8 and PGA20, respectively). The PCAs were performed using aggregations of trajectories with all the

three $t_{VST}$ conditions for each model. For each $t_{VST}$ condition, the ensemble calculated from the entire trajectory ($t_{run}$ = 50 ns and $N_{run}$ = 84) was considered as the reference ensemble. The FELs were compared with regard to $PCC_{PMF}$, similar to the Trp-cage case.

To assess the effects of $N_{run}$ and $t_{run}$, $PCC_{PMF}$ (and $P_{native}$ for the Trp-cage model) were calculated for ensembles with subsets of the reference trajectories. Because there are many possibilities to pick $N_{run}$ runs from 84 runs and $t_{run}$-length trajectories from the entire set of trajectories, we analyzed them by using the bootstrap approach. We constructed an ensemble by taking a random sampling of $N_{run}$ runs from 84 runs with replacement and repeated it 100 times. The statistics over the 100 ensembles were analyzed via simulation with this $N_{run}$ setting. This process was repeatedly performed for $N_{run}$ = 1, 2, …, 84. For the case of $t_{run}$, the trajectories were split into 5-ns bins, and an ensemble was constructed by taking a random sampling of $t_{run}/5$ bins with replacement. We confirmed that the results of the bootstrap analyses with 100 and 200 samples were consistent (Fig. S3).

The sampling efficiency was measured in terms of the frequency of traversals between low- and high-energy regions, which were defined as the ranges ($E_{min}$, $E_{low}$) and ($E_{high}$, $E_{max}$), respectively. Here, $E_{min}$ and $E_{max}$ denote the minimum and maximum potential energies in all the trajectories, respectively, and $E_{low}$ and $E_{high}$ are defined as follows.

$$E_{low} = E_{min} + X(E_{max} - E_{min}) \tag{6}$$

$$E_{high} = E_{max} - X(E_{max} - E_{min}) \tag{7}$$

$X$ is an arbitrary parameter in the range of 0–0.5. We assessed $X$ = 0.2 and 0.3. The traversal frequency $F_{travers}^{E}$ was calculated as the number of traversals between the two energy regions during 1.0 ns. The traversal frequencies of $RMSD_{native}$ and $R_g$ ($F_{travers}^{RMSD}$ and $F_{travers}^{Rg}$ respectively) were also analyzed.

## RESULTS

In the first part of this section, the results of the Trp-cage model are described. The reference ensemble is characterized in the subsection, "FEL of folding–unfolding equilibrium of Trp-cage." Next, the effects of the parameters $t_{run}$, $N_{run}$, and their balances are discussed in the successive subsections: "Effects of simulation time for each run," "Effects of number of independent runs," and "Balance between simulation time and number of runs," respectively. Subsequently, the effects of the other parameter $t_{VST}$ are discussed in the subsection, "Effects of frequency of bias-strength replacement." Additionally, the following subsection, "Effects of system complexity" describes the results of the PGA8 and PGA20 models and compares them with those of the Trp-cage model.

### FEL of folding–unfolding equilibrium of Trp-cage

For the Trp-cage model, we performed 34 iterations of TTP-V-McMD simulations while updating the estimation of the density of states, n($E$), and obtained a near-uniform energy distribution (Fig. S4). On the basis of this estimation, we performed production runs with $N_{run}$ = 84, $t_{run}$ = 200 ns, and $t_{VST}$ = 0.2 ps. This is called the reference setting

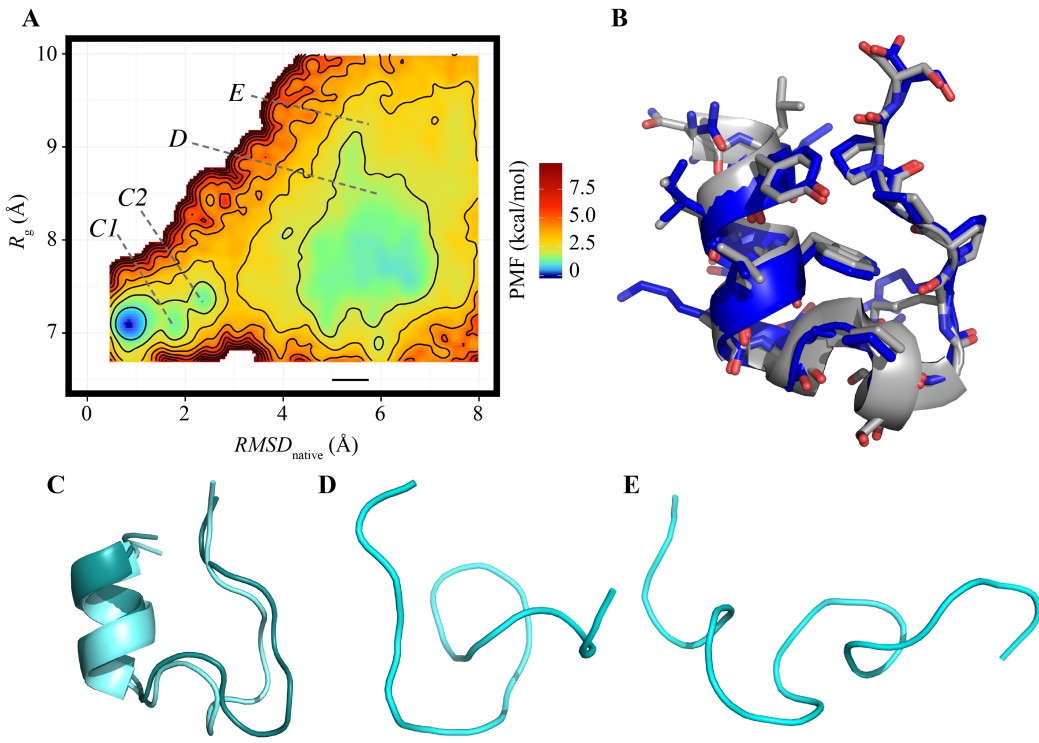

**Figure 1 FEL calculated by the reference ensemble of Trp-cage.** (A) FEL based on the RMSD$_{native}$–$R_g$ plane. The color gradation indicates the PMF. (B) Snapshot taken from the first basin (blue) superimposed on the experimentally solved structure (gray; PDB ID: 1Y2L, model 1). (C) Examples of snapshots near the first basin. The structures colored dark cyan and light cyan correspond to the positions C1 and C2 marked in (A), respectively. (D and E) Examples of unfolded structures in the second basin. The positions of each snapshot on the FEL are marked in (A).

hereinafter. The resultant canonical ensemble reweighted at 300 K is referred to as the reference ensemble.

The FEL of the reference ensemble projected on the RMSD$_{native}$–$R_g$ plane is shown in Fig. 1A. The most stable basin corresponds to the native structure consisting of an α-helix at the N-terminus, a 3$_{10}$-helix at the middle, and a loop region at the C-terminus (the secondary structural elements were recognized by using the DSSP software) (*Kabsch & Sander, 1983*). For example, the RMSD$_{native}$ of one of the most probable structures in this basin was 0.994 Å (Fig. 1B). The energy barrier (approximately 3.3 kcal/mol) was observed at RMSD$_{native}$ ≈ 3 Å in a low-$R_g$ regime. Around this barrier, the 3$_{10}$-helix at the middle of the peptide chain was partially deformed; this deformation can be the first step of an unfolding process (Fig. 1C). The details of the unfolding pathway are not discussed in this paper. The second basin was widely spread around RMSD$_{native}$ = 4–7 Å and $R_g$ = 7–9 Å. This corresponds to the unfolded state, and examples of the unfolded structures taken from this basin are shown in Figs. 1E and 1F. The difference in the PMF between the bottoms of the first and second stable basins was 1.014 kcal/mol, and the population of the native conformations ($P_{native}$) was 22.37%. The landscape is qualitatively similar to that calculated using the REMD method reported by another group

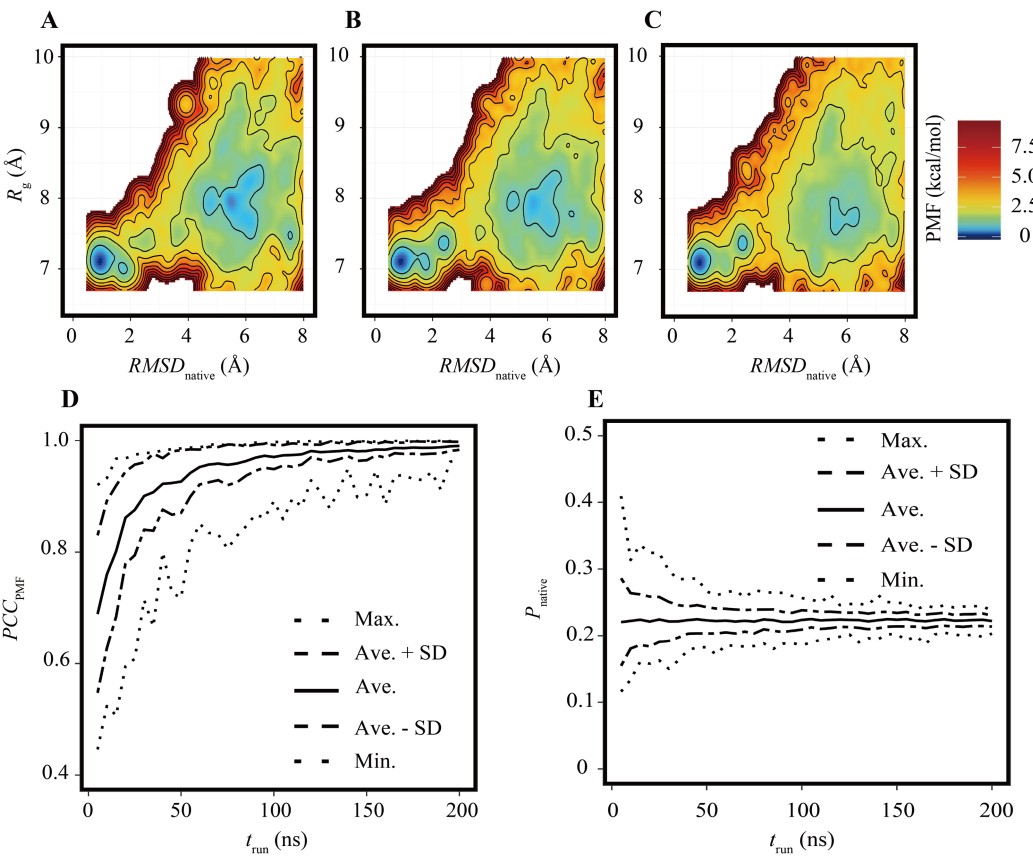

**Figure 2** **FELs of Trp-cage for various $t_{run}$ values with $N_{run} = 84$.** (A–C) FELs based on the trajectories of 0–25 ns (A), 0–50 ns (B), and 0–100 ns (C). (D) Bootstrap statistics of $PCC_{PMF}$. The solid line is the average, the dashed lines are the sum of the average and SD and the subtraction of the SD from average. The dotted lines indicate the maximum and minimum values among 100 randomly generated ensembles in each condition. (E) Statistics of $P_{native}$ shown in the same scheme as (D).

(*Day, Paschek & Garcia, 2010*). Our TTP-V-McMD simulation successfully identified the native structure as the most stable basin in the energy landscape, by using the reference setting.

## Effects of simulation time for each run

The FELs of the Trp-cage model were drawn for a variety of $t_{run}$ values under the condition of $N_{run} = 84$ and compared with the reference FEL. The FELs based on the trajectories of 0–25, 0–50, and 0–100 ns are shown in Figs. 2A–2C, respectively. The overall geometries of these FELs were qualitatively similar to the reference (Fig. 1A); their $PCC_{PMF}$ values were 0.936, 0.936, and 0.994, respectively. The bootstrap statistics of $PCC_{PMF}$ for each $t_{run}$ value are summarized in Fig. 2D. For $t_{run} = 200$ ns, the bootstrap average and the standard deviation (SD) of $PCC_{PMF}$ were 0.990 and 0.007, respectively. Even in the worst case among 100 randomly generated ensembles with $t_{run} = 200$ ns, $PCC_{PMF}$ was 0.966. From this condition, a decrease in $t_{run}$ yielded a slow decay of $PCC_{PMF}$, and $PCC_{PMF}$ reached 0.9 at $t_{run} \approx 30$ ns, which corresponds to 15% of the samples in the

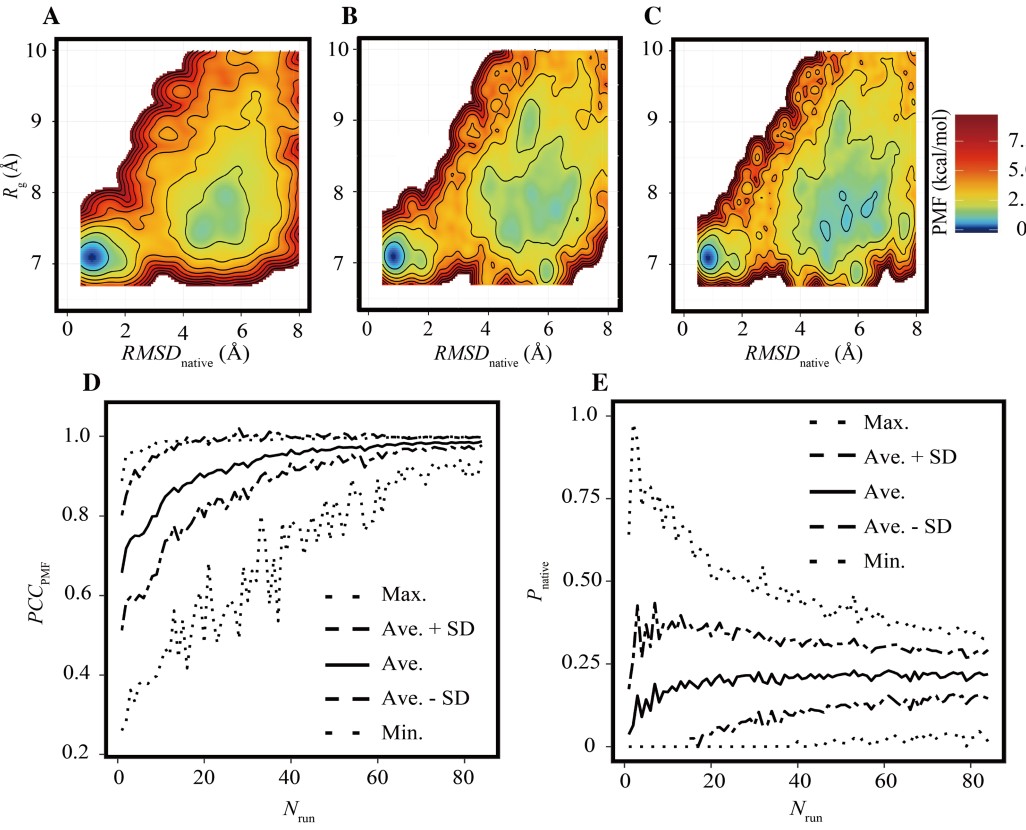

**Figure 3 Characteristics of FELs of Trp-cage for smaller $N_{run}$ values with $t_{run}$ = 200 ns.** (A–C) Examples of FELs with $N_{run}$ = 10 (A), $N_{run}$ = 21 (B), and $N_{run}$ = 42 (C). (D and E) Bootstrap statistics of $PCC_{PMF}$ (D) and $P_{native}$ (E). See also the legend of Fig. 2.

reference. Further decreasing $t_{run}$ resulted in a steep decrease of $PCC_{PMF}$. Along with the decrease of the bootstrap average of $PCC_{PMF}$, the SD was increased. This means that an insufficient simulation time causes a loss of robustness of the results.

In contrast to the fact that the $PCC_{PMF}$ decays in a shorter $t_{run}$ than the reference, the balance between the folded and unfolded states ($P_{native}$) was almost constant regardless of $t_{run}$ (Fig. 2E); the bootstrap average of $P_{native}$ for $t_{run}$ = 5–200 ns was in the range of 0.220 to 0.225. However, the SD of $P_{native}$ was reduced with the increase of $t_{run}$; the SDs of $P_{native}$ at $t_{run}$ = 5, 50, and 200 ns were 0.07, 0.02, and 0.008, respectively. The loss of robustness due to the insufficiency of the simulation time is demonstrated in terms of not only the similarity of the entire FEL but also the stability of the native fold.

### Effects of number of independent runs

As in the previous subsection, the effects of the reduction of $N_{run}$ on the FELs were assessed under the condition of $t_{run}$ = 200 ns. Examples of FELs with $N_{run}$ = 10, 21, and 42 are shown in Figs. 3A–3C, respectively; the $PCC_{PMF}$ values were 0.637, 0.939, and 0.993, respectively. Although the positions and wideness of the basins were similar to the reference, the FELs with a smaller $N_{run}$ were smoother and lacked small bumps on the landscapes. The bootstrap statistics of $PCC_{PMF}$ for various $N_{run}$ values (Fig. 3D) were

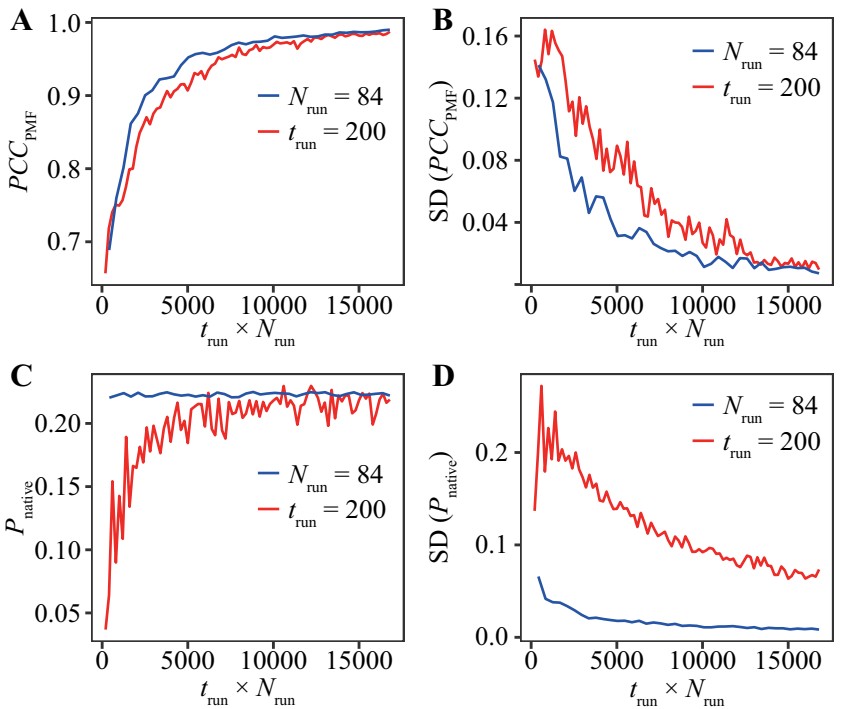

**Figure 4 Direct comparison between reducing $t_{run}$ with the fixed-$N_{run}$ condition (blue line) and reducing $N_{run}$ with the fixed-$t_{run}$ condition (red line) for the Trp-cage model.** The vertical axes indicate (A) the average of $PCC_{PMF}$, (B) the SD of $PCC_{PMF}$, (C) the average of $P_{native}$, and (D) the SD of $P_{native}$. The horizontal axis indicates the accumulated simulation length ($N_{run} \times t_{run}$).

similar to those for $t_{run}$ (Fig. 2D). The quantity of the samples required for $PCC_{PMF} \geq 0.9$ was approximately one-fourth of the reference ($N_{run} \approx 21$). The average (and the SD) of $PCC_{PMF}$ at $N_{run} = 21$ and 42 were 0.906 (0.07) and 0.956 (0.04), respectively. Larger $N_{run}$ values are needed to obtain robust results.

Regarding $P_{native}$, the influence of the reduction of $N_{run}$ (Fig. 3E) differed from that of the reduction of $t_{run}$ (Fig. 2E). A lower $N_{run}$ resulted in the underestimation of the population of native conformations. $P_{native}$ reached at plateau for $N_{run} \geq 21$. A certain number of runs was needed to obtain robust results, and $t_{run} = 200$ ns was too short to reach equilibrium with a small number of trajectories for this system.

## Balance between simulation time and number of runs

The evaluation for various $t_{run}$ values with $N_{run} = 84$ runs (Fig. 2) and that for various $N_{run}$ values with $t_{run} = 200$ ns (Fig. 3) indicate that reducing $t_{run}$ produced better results than reducing $N_{run}$ if the cumulative simulation time ($N_{run} \times t_{run}$) was the same. Figure 4 shows direct comparisons of the results, indicating that high-$N_{run}$ conditions resulted in a higher $PCC_{PMF}$ and more similar values of $P_{native}$ to the reference, with lower SDs, than long-$t_{run}$ conditions. In particular, the qualitative difference between the two strategies is shown by the mean of $P_{native}$. Reducing $N_{run}$ resulted in the significant underestimation of the fold stability, but reducing $t_{run}$ did not.
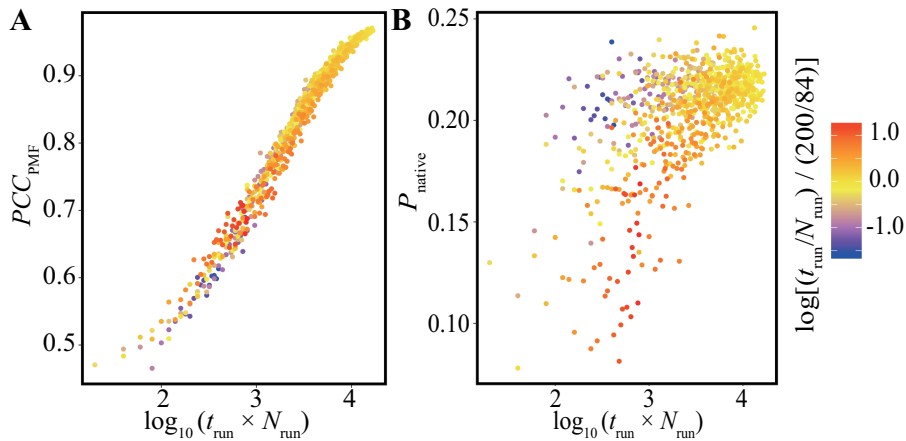

**Figure 5** Distribution of (A) the average of $PCC_{PMF}$ and (B) $P_{native}$ along the logarithm of the accumulated simulation length for various combinations of $N_{run}$ and $t_{run}$ extracted from the trajectories of the Trp-cage model. The color of each plot indicates the log-ratio of $N_{run}$ to $t_{run}$ compared with the reference. The definition is $\log[(t_{run}/N_{run})/(200/84)]$. This value becomes greater than 0 for conditions with a higher ratio of $t_{run}$ to $N_{run}$ than the reference.

In addition, we performed bootstrap analyses for all the combinations of 40-$t_{run}$ settings (5, 10, 15, …, 200 ns) and 21-$N_{run}$ settings (4, 8, 12, …, 84). The average values of $PCC_{PMF}$ and $P_{native}$ in all the conditions are presented in Fig. 5 and Fig. S5. The $PCC_{PMF}$ was proportional to $\log(N_{run} \times t_{run})$. While the trend of $P_{native}$ is ambiguous, the use of a larger number of samples resulted in a higher $P_{native}$. In the case where only small amount of data was available, a lower ratio of $t_{run}/N_{run}$ (purple plots in Fig. 5) yielded better results.

## Effects of frequency of bias-strength replacement

The parameter $t_{VST}$ controls the frequency of the bias-strength switching in the TTP-V-McMD method. We investigated the effects of this parameter by comparing the TTP-V-McMD simulations of the Trp-cage model under the three conditions—$t_{VST} = 0.002, 0.2$, and 20 ps—with $t_{run} = 50$ ns for $N_{run} = 84$.

Table 1 summarizes the frequency of traversals between high- and low-potential energy regimes ($F_{trv}^{E}$), as defined in Eqs. (6) and (7) with $X = 0.3$ and 0.2, as well as the frequency of traversals between $RMSD_{native}$ ($F_{trv}^{RMSD}$) and $R_g$ ($F_{trv}^{Rg}$). The simulations with a shorter $t_{VST}$ resulted in faster traversals in the potential energy space, indicating that with a shorter $t_{VST}$, a wider potential energy range can be sampled in a shorter time. However, faster traversal in the potential energy space does not ensure faster transition of the protein conformation. For both $X = 0.2$ and 0.3, although the setting of $t_{VST} = 0.002$ ps yielded the highest $F_{trv}^{E}$, this condition did not yield a higher $F_{trv}^{RMSD}$ and $F_{trv}^{Rg}$ compared to when a longer $t_{VST}$ was used. This result indicates that the relaxation of the conformation requires a longer time than that of the potential energy. If a strong bias is applied and the system takes a high-potential energy state, it can return to low-energy states before conformational changes. Therefore, a moderate speed for traversals in the potential energy space is ideal for efficient conformational sampling. In the case of $X = 0.2$, $t_{VST} = 0.2$ ps exhibited the most frequent conformational changes.

**Table 1** Average values (and the standard errors) of the traversal frequencies over 84 runs for the Trp-cage model.

| $t_{VST}$ (ps) | 0.002 | 0.2 | 20 |
|---|---|---|---|
| $X$ | | 0.3 | |
| $F_{trv}^E$ (ns$^{-1}$) | 1.63 (0.06) | 1.45 (0.04) | 1.02 (0.04) |
| $F_{trv}^{RMSD}$ (ns$^{-1}$) | 0.057 (0.006) | 0.060 (0.004) | 0.062 (0.006) |
| $F_{trv}^{Rg}$ (ns$^{-1}$) | 0.040 (0.005) | 0.044 (0.003) | 0.050 (0.006) |
| $X$ | | 0.2 | |
| $F_{trv}^E$ (ns$^{-1}$) | 0.70 (0.03) | 0.62 (0.02) | 0.46 (0.02) |
| $F_{trv}^{RMSD}$ (ns$^{-1}$) | 0.005 (0.001) | 0.011 (0.001) | 0.006 (0.002) |
| $F_{trv}^{Rg}$ (ns$^{-1}$) | 0.008 (0.002) | 0.012 (0.001) | 0.007 (0.002) |

In addition, the resultant ensembles were slightly affected by the setting of $t_{VST}$. We analyzed $P_{native}$ for ensembles of various $t_{run}$ values with $N_{run} = 84$ using the bootstrap method (Fig. S6). The results for all three $t_{VST}$ values showed similar trends, i.e., near-constant average values and the gradual decay of the SD with the increase of $t_{run}$. While $t_{VST} = 0.2$ ps showed a smaller $P_{native}$ than the other two $t_{VST}$ settings, the difference was smaller than the SD. On the other hand, higher SD values were observed in the following order: $t_{VST} = 0.2 > 20 > 0.002$ ps. This is consistent with the order of $F_{trv}^{RMSD}$ and $F_{trv}^{Rg}$ (Table 1). The result indicates that more frequent traversals between high- and low-RMSD$_{native}$ conformations make it possible to explore a wider region of the conformational space; thus, the population of the native conformation decreases, and the SD increases.

Regarding the $PCC_{PMF}$ with the reference setting ($t_{VST} = 0.2$ ps, $N_{run} = 84$, and $t_{run} = 200$ ns), the average $PCC_{PMF}$ values at $t_{run} = 50$ ns differed among different settings of $t_{VST}$ (Fig. S6). This indicates that changing $t_{VST}$ yields subtle differences in the resultant ensemble. Regarding the balance between $t_{run}$ and $N_{run}$, the trends were similar for all the settings of $t_{VST}$ (Fig. S7).

## Effects of system complexity: comparison with the PGA models

We performed the same analyses for the molecular models of PGA8 and PGA20. In contrast to Trp-cage, these peptides did not exhibit a particular fold. The FELs of both PGA8 and PGA20 were unimodal distributions, the basins of which consisted of a variety of collapsed conformations (Fig. 6 for $t_{VST} = 0.2$ ps). The ensembles included short secondary structural elements but they were unstable. Although the shape of the small bumps in the basins differed depending on the simulation conditions, the overall geometries of the FELs were similar (Fig. S8 for $t_{VST} = 0.002$ ps and 20 ps).

Regarding the balance between $t_{run}$ and $N_{run}$, Fig. 7 shows the bootstrap averages of $PCC_{PMF}$ between the ensemble calculated by the full-length trajectory ($t_{run} = 50$ ns and $N_{run} = 84$) and those calculated by the reduced trajectories. No clear differences were found between the $PCC_{PMF}$ curve with reduced $t_{run}$ and that with reduced $N_{run}$ for both the PGA8 and PGA20 (Fig. 7 for $t_{VST} = 0.2$ ps; Fig. S9 for the other conditions). A small number

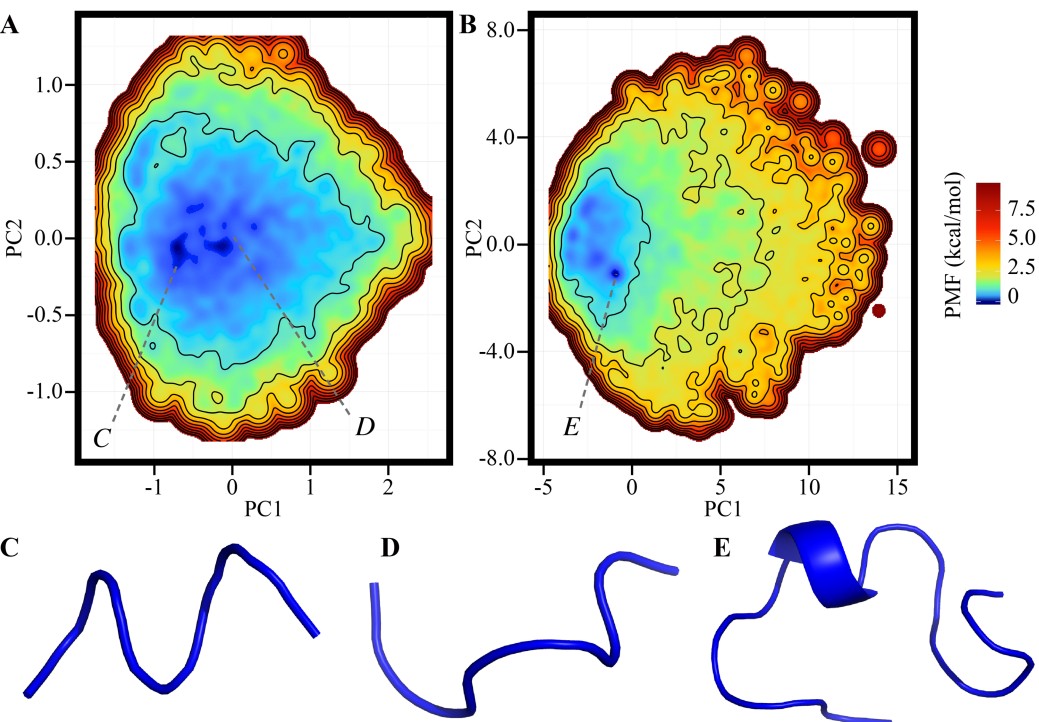

**Figure 6** FELs calculated by ensembles of (A) PGA8 and (B) PGA20 using $t_{\text{run}} = 50$ ns and $N_{\text{run}} = 84$ with $t_{\text{VST}} = 0.2$ ps. (C–E) Examples of snapshots in the basins marked in (A) and (B).

of long simulations exhibited the similar efficiency as that of many short simulations. In addition, no significant differences were found between the results of PGA8 and PGA20. It is noteworthy that the conformational space of PGA20 is considerably wider than that of PGA8 and similar to that of Trp-cage, because the conformational space volume of polypeptides is determined primarily by their length. Therefore, we concluded that the effects of balance between $t_{\text{run}}$ and $N_{\text{run}}$ are determined by the complexity of the FEL (e.g., existence of the free-energy barrier) rather than the conformational space volume. An increase in the number of runs is more effective for a system with more complex FEL.

For the PGA models, the frequencies of traversals in the potential energy and $R_{\text{g}}$ spaces ($F_{\text{trv}}^{E}$ and $F_{\text{trv}}^{Rg}$, respectively) are summarized in Table 2. Both the PGA8 and PGA20 models yielded similar trends as the Trp-cage model (Table 1). Although frequent replacements of bias-potential strength enhanced the traversals in the potential energy space, they did not enhance the conformational changes in terms of $R_{\text{g}}$. This implies that the conformational changes are much slower than the potential energy changes even if there is no free-energy barrier exists in the landscape. However, in contrast to the Trp-cage case, the drawback of the frequent replacement, that is, slow traversals in the conformational space, is unclear in the case of PGA20.

## DISCUSSION

We examined the performance of the TTP-V-McMD method with regard to two adjustable settings: (i) the balance between the number of runs ($N_{\text{run}}$) and the simulation

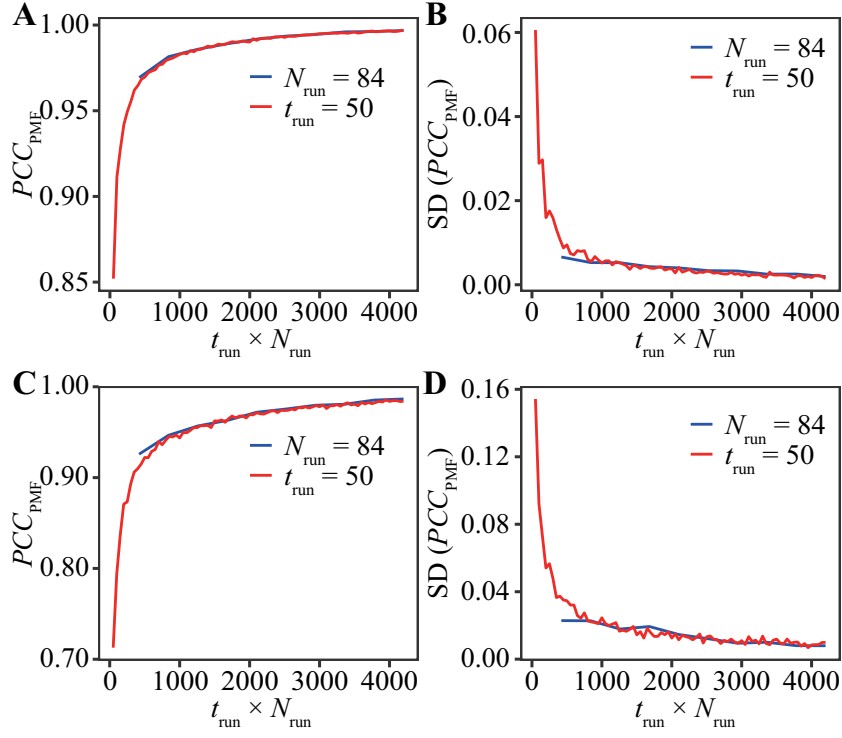

**Figure 7 Direct comparisons between reducing $t_{run}$ with fixed-$N_{run}$ (blue line) and reducing $N_{run}$ with fixed-$t_{run}$ (red line) for (A and B) the PGA8 and (C and D) PGA20 systems.** The vertical axes indicate (A and C) the bootstrap average of $PCC_{PMF}$, and (B and D) the SD of $PCC_{PMF}$. The horizontal axis indicates the accumulated simulation length ($N_{run} \times t_{run}$). The results of $t_{VST} = 0.2$ ps are presented. See also Fig. S9 for the other $t_{VST}$ conditions. 

**Table 2 Average values (and standard errors) of the traversal frequencies over 84 runs for PGA models.**

| Model | PGA8 | | | PGA20 | | |
|---|---|---|---|---|---|---|
| $t_{VST}$ (ps) | 0.002 | 0.2 | 20 | 0.002 | 0.2 | 20 |
| $X$ | 0.3 | | | 0.3 | | |
| $F_{trv}^{E}$ (ns$^{-1}$) | 2.91 (0.03) | 2.70 (0.03) | 0.86 (0.02) | 1.05 (0.04) | 0.99 (0.03) | 0.46 (0.02) |
| $F_{trv}^{Rg}$ (ns$^{-1}$) | 0.44 (0.02) | 0.47 (0.02) | 0.47 (0.02) | 0.045 (0.005) | 0.044 (0.005) | 0.049 (0.006) |
| $X$ | 0.2 | | | 0.2 | | |
| $F_{trv}^{E}$ (ns$^{-1}$) | 1.58 (0.02) | 1.53 (0.02) | 0.50 (0.01) | 0.4 (0.02) | 0.41 (0.01) | 0.22 (0.01) |
| $F_{trv}^{Rg}$ (ns$^{-1}$) | 0.117 (0.007) | 0.147 (0.007) | 0.146 (0.007) | 0.011 (0.002) | 0.013 (0.002) | 0.015 (0.003) |

length in each run ($t_{run}$) and (ii) the frequency of the bias-strength replacement ($t_{VST}$). For (i), in the Trp-cage model including folding–unfolding transition, we found higher robustness of the conditions with a larger number of runs than with longer simulations. In particular, the probability of the existence of native conformations in a resultant ensemble ($P_{native}$) was more sensitive to the condition than the entire similarity of the FEL. However, for the cases of PGAs without free-energy barrier in their FELs, no significant effect was shown in the balance between the number and length of simulations. Therefore, the optimal balance depended on the molecular system, and the complexity of

the FELs was a key feature rather than the degree of freedom. In any case, increasing the number of simulations was recommended because it is not worse than increasing the length of each run. This result is practically useful because performing many parallel runs is easier than executing a single long simulation. While the result obtained here encourages performing many short runs, it requires the condition that the initial structures of the production runs are uniformly sampled from the multicanonical ensemble, whose energy distribution is uniform (*Ikebe et al., 2010*). As our protocol samples the initial structures of the production runs from the previous iteration of the McMD, it is expected that this condition holds.

For (ii), whereas higher frequencies of bias-strength replacement enhance the sampling of a wider range of potential energy, they do not ensure the enhancement of the sampling of a wider range of conformations. This means that the enhancement of the sampling along one variable (e.g., potential energy or temperature) does not ensure the enhancement of the sampling along another variable (e.g., RMSD and $R_\mathrm{g}$). Rapid traversals in the energy space sometimes obtain a high energy and return to the low-energy regime before conformational change regardless of the existence of free-energy barrier in the FEL. A moderate frequency is needed to maximize the performance for any molecular system.

The findings that we obtained by applying the TTP-V-McMD method provide insight into the characteristics of many other GE methods. (i) For GE methods that involve running independent parallel simulations, e.g., simulated tempering and AUS, performing many short runs can be more effective than increasing the length of each run. For GE methods where parallel runs are coupled, e.g., the REMD method, this conclusion should not be simply applied. For example, an increase in the number of runs in the REMD method resulted in larger overlaps of the distributions of neighboring replicas, along with an increase in the acceptance probability of replica-exchange trials. Our previous evaluation for the REMD method showed that a larger number of replicas does not always yield better results (*Iwai, Kasahara & Takahashi, 2018*). The number of runs should be adjusted independently from the coupling condition of the parallel runs; for example, the number of runs in a REMD simulation could be increased by performing two or more independent REMD simulations with different initial conformations, and aggregating the resultant ensembles. (ii) Regarding the frequency of the bias-strength replacement, the conclusion that the interval should be long enough to relax the conformation could be transferred to other GE methods. For the REMD methods, the effects of the interval for replica-exchange trials have been reported; while some studies recommended shorter intervals (*Sindhikara, Meng & Roitberg, 2008*; *Sindhikara, Emerson & Roitberg, 2010*), the side effects of highly frequent exchange trials have also been reported and were consistent to our result (*Periole & Mark, 2007*; *Iwai, Kasahara & Takahashi, 2018*).

## CONCLUSIONS

In this study, the effects of two parameters of GE methods, i.e., (i) the balance between the number of runs ($N_\mathrm{run}$) and the simulation length in each run ($t_\mathrm{run}$) and (ii) the frequency of the bias-strength switching ($t_\mathrm{VST}$) were extensively examined with using all-atom explicit-solvent models of three polypeptides that are a foldable mini-protein and disordered

peptides. We suggest a guide to adjust the setting for general molecular systems and GE methods. (i) Increasing in the number of runs should be prioritized rather than increasing the simulation length. (ii) Highly frequent replacements of the bias potentials may yield side effects because conformational relaxation was slower than potential energy relaxation. The time interval for replacement should be longer than or equal to 0.2 ps.

## ACKNOWLEDGEMENTS

The MD simulations were performed using the TSUBAME3 supercomputer at Tokyo Institute of Technology, provided through the HPCI System Research Projects (Project IDs: hp180050, hp180054, hp190017, and hp190018). The numerical computations for post-simulation analyses were performed using the supercomputer system provided by the National Institute of Genetics, Research Organization of Information and Systems, Japan.

### Funding

Kota Kasahara is supported by Japan Society for the Promotion of Science (JSPS) KAKENHI Grant No. JP16K18526. Junichi Higo is supported by JSPS KAKENHI Grant No. JP16K05517 and by the Development of core technologies for innovative drug development based upon IT from Japan Agency for Medical Research and Development, AMED. The funders had no role in study design, data collection and analysis, decision to publish, or preparation of the manuscript.

### Grant Disclosures

The following grant information was disclosed by the authors:
Kota Kasahara is supported by Japan Society for the Promotion of Science (JSPS) KAKENHI: JP16K18526.
JSPS KAKENHI: JP16K05517.
Development of core technologies for innovative drug development based upon IT from Japan Agency for Medical Research and Development, AMED.

### Competing Interests

The authors declare that they have no competing interests.

### Author Contributions

- Takuya Shimato performed the experiments, analyzed the data, contributed reagents/materials/analysis tools, prepared figures and/or tables, performed the computation work, approved the final draft.
- Kota Kasahara conceived and designed the experiments, performed the experiments, contributed reagents/materials/analysis tools, performed the computation work, authored or reviewed drafts of the paper, approved the final draft.
- Junichi Higo conceived and designed the experiments, authored or reviewed drafts of the paper, approved the final draft.

● Takuya Takahashi conceived and designed the experiments, authored or reviewed drafts of the paper, approved the final draft.

## Data Availability

The in-house MD simulation software named myPresto/omegagene is available at http://www.protein.osaka-u.ac.jp/rcsfp/pi/omegagene/.

The raw data of structural parameters of MD simulation trajectories are available at figshare: Kasahara, Kota (2019): fel.tar.gz. figshare. Journal contribution. DOI 10.6084/m9.figshare.8797895.v2.

## Supplemental Information

Supplemental information for this article can be found online at http://dx.doi.org/10.7717/peerj-pchem.4#supplemental-information.

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
