# Peer review of "Effects of number of parallel runs and frequency of bias-strength replacement in generalized ensemble molecular dynamics simulations"

_PeerJ Physical Chemistry, doi:10.7717/peerj-pchem.4_

## Round 0.1 · original submission · Minor Revisions

As you can see, the reviewers have overall quite positive comments on your manuscript, with some minor details to be revised. Please take into account, in particular, the comments of reviewer 1.

Reviewer 1 ·

Basic reporting

The authors have written clear and unambiguous sentences, which fully meet the criteria of this paper. However, three minor fixes are required as follows:

1. The context on lines 74-76 "It is reported that ....." doesn't have the reference. By clarifying who did the study, you can avoid the confusion of readers.

2. "0.002 ps ns," on line 205 is a spelling mistake. I think this is "0.002 ps,".

3. The simulation condition t_vst of the data shown in figure 6 is different between line 389 of the text and the caption. I guess that the text is correct.

Experimental design

no comment.

Validity of the findings

Research questions well defined and their conclusions are clearly described. I'm convinced that this paper provides useful information to researchers who are using simulation methods similar to this study.

Additional comments

I require two corrections to improve the reader's readability as follows:

1. The types of lines in the line graph as shown in figure S6 should be reconsidered. It is difficult for readers to distinguish between the two dotted lines in the caption.

2. There are too many digits in the numerical data shown in the text such as PCC_PMF and P_native. Especially, there is no point in displaying the value of standard deviation with many digits.

·

Basic reporting

I do not see any raw data in the SI. As much raw data as is practical should be provided.

Experimental design

no comment

Validity of the findings

no comment

Additional comments

The work appears to be carefully done, and the authors make clear recommendations based on an analysis of the data

Reviewer 3 ·

Basic reporting

The quality of writing is excellent through. The literature references are appropriate and sufficient. The figures look good and the paper is complete.

Experimental design

The authors report appropriate analysis of the efficiency of generalized ensemble simulations for protein folding. It answers some fundamental questions about the efficiency of these methods and the optimal simulation options for these models. The quality of the models is the norm for the field. The calculations are done rigorously and the methods are well-described.

Validity of the findings

The findings are valid and are consistent with their reported data. The results are presented honestly and unambiguously.

---

## Round 0.2 · accepted · Accept

Based on your revised manuscript I am happy to accept the paper.